# Conformity of Annual Reports to an Integrated Reporting Framework: ASE Listed Companies

**Ghada A. Altarawneh** [1,2,*] **and Asma'a Omar Al-Halalmeh** [1]

1   Accounting Department, Mutah University, Karak 61711, Jordan; galtarawneh@ut.edu.sa
2   Accounting Department, Faculty of Business Administration, University of Tabuk,
    Tabuk 47512, Saudi Arabia
*   Correspondence: ghadatrn@mutah.edu.jo; Tel.: +966-14-427-3022

**Abstract:** The objectives of this study are to determine the level of conformity between Current Issued Reports (CIRs) and Integrated Report (IR) elements of the Amman Stocks Exchange (ASE) listed companies, as well as to determine whether the investigated corporate characteristics (size, age, quality assurance (QA), earning per share (EPS), industry type, foreign ownership (FO)) of these companies have any impact on the conformability of CIRs. It is worth mentioning that (QA), and (EPS), have never been examined by looking at its association with corporate disclosures, and IR in particular. Based on adoption of the IR framework and using the method of content analysis, corporate annual reports and other stand-alone reports of 82 companies in 2017 and 2018 within the financial, industrial, and services sectors, were chosen for this study. The findings of the study provide an answer to the research question and show that sectors vary in their levels of conformity. It reveals that the service sector shows the lowest conformability compared to other sectors, whereas the financial firms conform 65%, followed by the industrial sector. It also finds a positive association between CIRs conformability and variables of size, age of company and quality assurance. However, EPS, FO and type of industry were found to have no impact on the conformability of CIRs to the IR framework. This study has contributed to IR research, which, as a field, has previously received very little recognition among scholars in Jordan. Moreover, IR still does not exist in Jordan's business practices.

**Keywords:** accounting disclosure; integrated reporting; international integrated reporting council; content elements; content analysis; ASE

**JEL Classification:** M41; M14; H83

## 1. Introduction

Corporate disclosure is significant for the reliability and efficiency of the capital market. Firms have different channels by which they disclose information, such as financial statements, corporate social responsibility reports, and management reports. A few years ago, the nature and needs of business changed dramatically in order to fulfill needs for market information. Stakeholders are aware that traditional approaches to corporate reporting have become inadequate for providing necessary information (Cohen et al. 2012). Issues such as a financial crisis, or scandals that spread distrust and economic uncertainty amongst users of financial reports (Adams and Simnett 2011), have changed the nature of information required, and have placed additional pressure on companies and standard setters to enhance the quality of corporate reporting. Information should reflect corporate transparency and accountability should be forward-looking, and not limited to backward-looking presentations (Beattie et al. 2004). This demand necessitates businesses to create innovative ways that take into consideration the new nature of information that companies must provide.

Based on this demand, many firms have started to improve their traditional methods of financial reporting by issuing nonfinancial information, either in the form of stand-alone sustainability reports, corporate social responsibility (CSR) reports or within their financial reporting (Simnett et al. 2009; KPMG 2011; Cohen et al. 2012). However, these supplementary disclosures, despite being relevant and useful for decision-makers, have been criticized for being reported in a way that does not assist users to understand, compare and assess current and future performances of companies. The material is often weighty, and contains low quality information "that stakeholders find difficult to assess in terms of veracity and completeness" (Siebecker 2009). These corporate reporting challenges have stressed the need for integrated reporting that will efficiently combine financial and nonfinancial information in a meaningful manner (IIRC 2011; Solomon and Maroun 2012).

In 2011, the debate about the future of corporate reporting reached a turning point when the International Integrated Reporting Council (IIRC) launched a global IIRC pilot to develop an integrated reporting framework. The IIRC is a "global coalition of regulators, investors, companies, standard setters, the accounting profession and NGOs[1]" (IIRC 2013), who share the view that communications of value should be the next step in the development of corporate reporting.

The IIRC proposed that Integrated Reporting (IR) enables firms to present and communicate material information about their strategies, governance, performance, different prospects and value creation over the short, medium, and long term (IIRC 2013, 2015) in a clear, concise and comparable manner. IR was introduced as an emerging accounting method to enable companies to understand how they create value and provide a full and effective picture to stakeholders. The key IR feature is combining a company's financial and nonfinancial information in one report.

In the field of contemporary accounting research, IR is considered a notable and growing topic. Several studies have been presented at prominent and leading accounting conferences. Research in IR is now beyond studying its relevance or raising awareness regarding its importance as a tool to meet stakeholder's needs for information (Eccles et al. 2015). Studies of IR recently focused on issues such as its effect on financial performance and value creation (Baboukardos and Rimmel 2016), and on the "fundamental concepts," "guiding principles" and "content elements" of an integrated report, and raised some issues concerning the "preparation and presentation" of such a report (IIRC 2013; Flower 2015; Adams 2015).

As described above, traditional and current reporting approaches do not meet the needs of modern society. Current reporting approaches are widely criticized for focusing too narrowly on the financial aspects of company performance (De Villiers et al. 2014). In addition, even the presentation of nonfinancial information makes it very difficult for investors and stakeholders to compare the performances of different companies (Eccles and Saltzman 2011). Therefore, a departure from traditional financially biased reporting is needed, (Ioannou and Serafeim 2015) towards an IR approach.

Practically, many companies worldwide have adopted IR (Dumay 2016; Dumay et al. 2016), yet traditional financial reporting and some voluntary environmental, social and governance reports dominate, or are the only reporting approach used in developing countries, where Jordan is no exception. Jordan has yet to adopt such a reporting system, thus, it would be useful to know where Jordanian companies stand with respect to IR. In this vein, this paper has two main questions:

- To what extent do Currently Issued Reports (CIRs) of AES listed companies conform to elements of the Integrated Reporting Framework?
- What is the impact of corporate characteristics (Size, EPS, quality assurance, foreign ownership, and sector) on the levels of conformity of CIRs to include IR elements in companies' reports?

The study makes a contribution in several ways; it is the first to approach this research within the context of Jordan. It serves as a starting point for future studies, and may contribute to changing the

---

[1] Nongovernmental Organization.

way companies disclose their information. In addition, this study makes a contribution by providing an initial assessment and practical vision on where CIRs of ASE stands with regard to IR requirements. It is worth mentioning that QA, and EPS, have never been examined by looking at its association with corporate disclosures, and IR in particular. This study contributes to the literature by responding to various recent calls in the area of IR. It also brings insight from developing countries, where little is known about this topic.

The remainder of the paper is organized as follows. Section 2 provides a background to the integrated report. Section 3 provides a brief explanation of the literature review and presents the hypotheses of the study. Section 4 outlines the overall research process and the research design, including research methods explains the research sample, methodology and disclosure index. Section 5 presents the results and discusses the findings, draws conclusions, highlights limitations in the study, makes recommendations and suggests areas for future research.

## 2. What Is the Integrated Reporting (IR)?

An integrated reporting is "A concise communication about how an organization's strategy, governance, performance and prospects, in the context of its external environment, lead to the creation of value over the short, medium and long term" (IIRC 2013, p. 7).

The integrated report aims to provide the companies' stakeholders with a holistic and comprehensive picture (Owen 2013) of the organization's different aspects, such as how its resources are creating value, future prospects, business model, strategies, risks, performance and sustainability, in a clear and concise way. Therefore, it encourages and supports integrated thinking that emphasizes the creation of value over the short, medium, and long term. A company that adopts Integrated Reporting would publish a report that wrapping all the financial, social, governance and environmental information together into an integrated format. The integrated reporting framework specifies its "guiding principles" and "content elements" as shown in Tables 1 and 2.

**Table 1.** The International Integrated Reporting Council (IIRC) guiding principles.

| Guiding Principle | Meaning |
|---|---|
| Strategic focus and future orientation | An IR should provide insight into the organization's strategy and how it relates to the organization's ability to create value in the short, medium and long term and to its use of and effects on the capitals |
| Connectivity of information | An IR should show a holistic picture of the combination, interrelatedness and dependencies among the factors that affect the organization's ability to create value over time |
| Stakeholder relationships | An IR should provide insight into the nature and quality of the organization's relationships with its key stakeholders, including how and to what extent the organization understands, takes into account and responds to their legitimate needs and interests |
| Materiality | An IR should disclose information about matters that substantively affect the organization's ability to create value over the short, medium and long term |
| Conciseness | An IR should be concise |
| Reliability and completeness | An IR should include all material matters, both positive and negative, in a balanced way and without material error |
| Consistency and comparability | The information in an integrated report should be presented: (a) on a basis that is consistent over time; and (b) in a way that enables comparison with other organizations to the extent that it is material to the organization's own ability to create value over time |

Source: Adapted from International Integrated Reporting Council (IIRC 2013).

**Table 2.** The IIRC content elements.

| Content element | Question to answer | Including |
|---|---|---|
| Organizational overview and External environment | What does the organization do and what are the circumstances under which the organization operates? | The organization's mission and vision; Key quantitative information; significant factors affecting the external environment and the organization's response |
| Governance | How does the organization's governance structure support its ability to create value in the short, medium and long term? | Organizations' leadership structure; specific process and particular actions; remuneration and incentives |
| Business model | What is the organization's business model? | Inputs, business activities; Outputs, outcomes |
| Risks and opportunities | What are the specific risks and opportunities that affect the organization's ability to create value over the short, medium and long term? and how is the organization dealing with them? | The specific source of risks and opportunities; the organization's assessment of risks; the specific steps taken to manage risks |
| Strategy and resource allocation | Where does the organization want to go and how does it intend to get there? | The organization's strategic objective; The resource allocation plan; The linkage between them |
| Performance | To what extent has the organization achieved its strategic objectives for the period and what are its outcomes in terms of effects on the capitals? | Quantitative indicators on targets and risks; the organization's effects on capitals; the state of key stakeholders' relationships; linkages with past and future performance |
| Outlook | What challenges and uncertainties are the organization likely to encounter in pursuing its strategy, and what are the potential implications for its business model and future performance? | The organization's expectations and how the organization is equipped to face them; the discussions of potential implications for future financial performance |
| preparation and presentation | How does the organization determine what matters to include in the integrated report and how are such matters quantified or evaluated? | The organization materiality process; the description of reporting boundary; frameworks and methods used to quantify or evaluate material matters |
| General reporting guidance | | Disclosure of material matters; disclosures about the capitals; time for short-, medium and long-term aggregation and disaggregation |

Source: Adapted from International Integrated Reporting Council (IIRC 2013).

Different studies such as (Ernst and Young 2014) pointed out that there are many benefits to adopt IR, such as the following:

1. enables companies to recognize and assess risks,
2. enhances decision making,
3. provides forward-looking information;
4. provides a comprehensive and concise overview;
5. boosts the importance and quality of governance
6. improves the organization's image
7. improves the relationships with stakeholders;
8. improves interdepartmental and capitals connections and relationships

## 3. Literature Review

The concept of integrated reporting (IR) has been proposed and researched academically and practically with the aim of developing corporate reporting and enhancing its usefulness and efficiency.

Many researchers have investigated issues of integrated reporting using different approaches. Some studies investigated the possible impact of corporate characteristics on IR adoption, such as the size of the company, profitability and the existence of a sustainability report. For instance, Kilic and Kuzey (2018a) investigated the adherence level of current companies' reports to the IR framework through the analysis of whether and to what extent those reports include the content elements of this framework. This study also aims to examine the impact of corporate sustainability characteristics on the adherence level of current company reports to the integrated reporting framework. The sample for research contains the nonfinancial companies, which were listed on Borsa Istanbul, as of 31 December 2015. The results show that current company reports mainly present generic risks, provide positive information while dismissing negative information, present financial and nonfinancial initiatives separately; lack a strategic focus, and include backwards-looking information rather than forward-looking information. Consistent with the predictions, the authors found that the IR is significantly and positively associated with sustainability reporting, Global Reporting Initiative (GRI) adoption, sustainability index listing, and the presence of a sustainability committee.

Further, Ali (2017) conducted a study to determine the extent of integrated reporting practices amongst 106 companies listed on the Saudi stock market (Tadawul) and investigate the factors that influence such practices over the period from 2013 to 2014. The sample comprises all of the nonfinancial companies listed on the Saudi stock market (Tadawul). The study developed an integrated reporting index comprised of 45 items. The analysis of the reports shows that the extent of IR practices is still limited with little improvement evidenced throughout the investigated period. The study found a significant association between IR practices and size and auditor type in both years. Insignificant results were reported regarding profitability and industry type.

Akhter and Ishihara (2018) examined IR of some early adopting companies of the UK. The contents of integrated reports of five selected companies are assessed against a disclosure checklist based on the IR Framework. The results show that the disclosure rates vary from 51 percent to 70 percent. This range represents a moderate level of compliance in a regulatory environment where preparation of integrated reports, as per the IIRC, is not mandatory. On the other hand, a small amount of information was disclosed in some areas such as, future-outlook, opportunities, or material issues. In general, the reports lack connectivity in varying degrees.

Other researchers examined the gap between what IR requires and what corporate companies disclose. For instance, Stent and Dowler (2015) carried out a study to provide early assessments of the changes for corporate reporting processes. The researchers developed a reporting checklist based on the requirements for IR, which they use to assess the gap between current "best practice" reporting processes and IR. The study evaluates 2011 annual reports and related online reporting practices for four New Zealand "best practice reporting entities" using their reporting checklist. Although none of the sample entities published a fully-integrated report for 2011, reporting scores the range from 70 to 87 percent.

Some of researchers have tried to capture the perception of different stakeholders and decision makers towards the adoption of IR (Perego et al. 2016; Rowbottom and Locke 2014) and comprehend challenges and insufficiencies in integrated reporting (De Villiers et al. 2014; Adams 2015; Flower 2015).

For instance, Anojan (2019) examined the perception of accounting experts on the implementation and limitation of integrated financial reporting in Sri Lanka, and the appropriate way to encourage integrated financial reporting in Sri Lanka. The result found that the opportunities and benefits of the implementation of integrated reporting are more than challenges and there is a lack of knowledge and awareness regarding integrated reporting in Sri Lanka.

Hassan (2017) conducted an empirical study of the readiness of the Egyptian capital market to move to the mandatory application of integrated reports. The most remarkable steps are the development of

the company responsibility index, as well as the issuance of a number of strict requirements and rules for the purpose of enhancing disclosure and transparency. A content analysis was used to determine the level of actual disclosure of Egyptian companies, and identify the determinants of integrated disclosure. The study found that the Egyptian capital market is moving towards integrated reporting. Thus, effort has to be made by organizational and professional authorities and other stakeholders in order to exercise pressure on companies to accelerate the transformation process. On the other hand, results provide evidence that there is a positive relationship between company listing on S&P EGX index and integrated disclosure level, as well between board size and auditing the company by one of the big four audit firms, and integrated disclosure level. Accordingly, the results reveal that company characteristics and corporate governance structures are major determinates for integrated level.

Naynar et al. (2018) explored the emphasis placed on certain integrated reporting themes by financial services companies and stakeholders' perception of the importance of these themes to ascertain if a perception gap exists. The study also considers if the perception gap is affected by user sophistication. The results of this study explained that a perception gap existed because companies do not fully understand what information is valued by their stakeholders. In addition, the study demonstrates that sophistication has an effect on the type of disclosures that are valued by users and the manner in which the disclosures are presented.

Perez (2018) examined whether the quality of IR disclosures, the assurance of sustainability performance, the use of assurance standards the international standard on assurance engagements (ISAE3000) and Assurance Standard (AA1000AS), and the level of information audited are all associated with market liquidity and lower analyst forecast error. The researcher analyzed the best available selection of companies listed on the Johannesburg Stock Exchange in South Africa, from 2013 to 2015. The major factor driving the selection of this particular period was to analyze not only existing IR practice but also to investigate IR two years after King III came into force, when firms had time to develop a mature response to the changed reporting environment. Because IR became mandatory on an "apply or explain" basis for listed JSE firms from 2010. The results show that IR quality is associated with lower analyst error and positively associated with market liquidity. The evidence also indicates that the earnings forecast error is lower for firms in the materials sector of the South African economy. Forecast errors are higher for companies with volatile returns and lower for larger firms, which is consistent with prior research. Contrary to expectations, the assurance of nonfinancial information in IR does not have a significant effect on analyst forecast accuracy. These results suggest that, in a setting such as South Africa, the assurance of sustainability performance does not provide additional informative value to analysts, irrespective of who provides the assurance and of the level of information. In contrast, the assurance of sustainability disclosures is associated with market liquidity. Similar results are found for those companies that use assurance standards. Overall, these findings support the advantages of IR, thus providing useful information to capital markets. Additionally, this evidence progresses the discussion on the economic incentives necessary to assure nonfinancial information.

Kilic and Kuzey (2018b) examined the nature, determinants, and extent of forward-looking disclosures in early examples of integrated reporting. The forward-looking disclosure index (FLDI) was categorized into two main groups, quantitative and qualitative, including 30 items in total. Contrary to the researcher's expectation, the results show that the majority of the entities tended to provide qualitative forward-looking disclosures rather than quantitative. Further, the findings showed that gender diversity and firm size are positively related to forward-looking disclosures, whereas leverage is negatively related to forward-looking disclosures. Contrary to expectations, the researchers did not find a significant impact of board size, board composition, profitability, or industry on forward-looking disclosures.

As mentioned, our research investigated the conformity level of current company reports of ASE to the IR framework through analyzing whether or to what extent those reports included the content elements of this framework. Therefore, we measured the integrated reporting score of each company via a manual content analysis of its annual reports and stand-alone sustainability reports.

In addition, our research examined the impact of corporate characteristics on the level of conformability of current company reports to the IR framework. Our research extends the findings of Stent and Dowler (2015) and Kilic and Kuzey (2018a) through examination of the developing country case with a larger sample size, covering large, medium and small sized firms to ensure diversity between companies of our sample.

The sample of our study is significantly larger than those in previous studies (Casonato et al. 2018; Silvestri et al. 2017; Stent and Dowler 2015). Furthermore, unlike previous studies, which are limited to a certain business sector (Camodeca et al. 2019; Alqallaf and Alareeni 2018; Naynar et al. 2018) the scope of the current study extended to involve various sectors including (Banks, Insurance, Manufacturing and Services). It is worth mentioning that QA, as one of the variables investigated in this study, has never been examined by looking at its association with IR in particular.

Taking into consideration IR previous literature, and in order to answer the research question, we propose to test the following two main hypotheses:

**Hypothesis 1 (H1).** *There are no significant statistical differences between the level of conformability of CIRs to include content elements of the IR framework related to corporate characteristics (QA, FO, and sector).*

**Hypothesis 2 (H2).** *There is no significant statistical impact for corporate characteristics (size, EPS, QA, FO, age, and sector) on the level of conformability of CIR to include content elements of the IR framework.*

## 4. Research Methodology and Design

The most feasible and therefore suitable approach is to analyze the content of corporate annual reports (CARs) and other available reports, which were chosen due to accessibility and reliability (Bell et al. 2018). Content analysis used as a suitable method to examine the selected CIRs. A most commonly used form of content analysis is to analyze the existence or absence of each item (Krippendorff 2004). This approach has been used in many prior studies (Oliveira et al. 2010; Frías-Aceituno et al. 2013; García-Sánchez et al. 2013; Setia et al. 2015; Haji and Anifowose 2016, 2017). Thus, it allows for drawing certain inferences from the documents by systematically focusing on CIR content in order to identify elements of IR as well as investigating characteristics within the data. We hope to reveal similarities between what companies report and what is required by the IR framework. In addition, this study will perform an analytical analysis to detect the impact of a set of characteristic variables on the conformability of company annual reports.

Content analysis is "a technique for gathering and analyzing the content of text. The content refers to words, meanings, pictures, symbols, ideas, themes or any message that can be communicated" (Neuman 2003, p. 219). This technique has been widely used in social and environmental disclosures in order to find the subjects covered in sustainability reports (Guthrie and Farneti 2008). In addition, it is used to rank firms that report sustainability by verifying if sustainability information and disclosed indexes match the items proposed by international reporting standards such as GRI guidelines (Tewari and Dave 2012).

Moreover, comparable to the approach used by different researchers, (Stent and Dowler 2015; Marx and Mohammadali-Haji 2014; Lee and Yeo 2016; Frías-Aceituno et al. 2013; Setia et al. 2015) we applied a disclosure index covering 41 elements of the content elements of the IR framework to determine the integrated reporting disclosure score (IRS) of the selected companies. The employed disclosure index included 41 elements within seven categories.

In the first part of the current research, the study uses multiple performance descriptors to assign weights for the IR elements checklist (i.e., 0, 1, 2 or 3 depends on the item). Assigned weights help to disclose the extent to which a certain company confirms its similarity to IR index. In the second part, we apply a descriptive analysis to determine the compliance of the sample companies with the elements of IR. Finally, we performed an analytical analysis to detect the impact of a set of demographics (firm)

characteristics on the company conformability of corporate reports and the corresponding elements of IR. Figure 1 shows our approach.

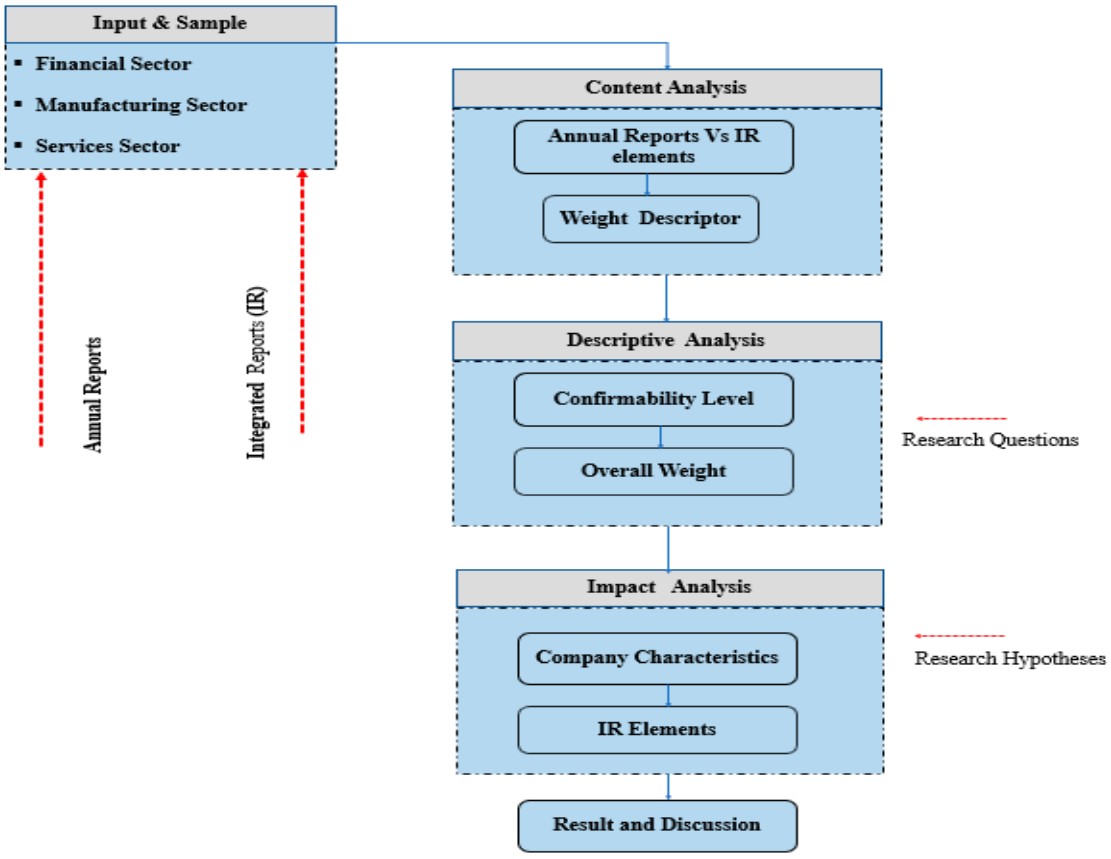

**Figure 1.** Research Approach.

### 4.1. Input and Sample

Secondary resources are used to collect the required data. The current issued reports of the selected listed firms in years (2017) and (2018) were identified using the website of Amman Stocks Exchange (supplementary materials, https://www.ase.com.jo/en). All data obtained from ASE online platforms were downloaded onto either Excel or Adobe PDF file. The corporate reports are a reliable and valid source for examination (Bell et al. 2018). The sample consists of 86 companies from main industrial sectors ((financial sector including banking and insurance firms that represent the whole finance sector in Jordan), manufacturing, and service sector), taking into consideration that the selected companies have represented large, small and medium-sized enterprises. However, deductions of companies with missing data for one or more variables for the period of the study brought the final sample to be 82 companies[2] as shown in Table 3. This resulted in 164 traditional financial reports, and then we added other available reports and stand-alone reports to make up the entire sample for analysis. The sample was examined by using 41 items, each of which is related to a specific IR element.

Moreover, the characteristics of the selected companies in terms of size, Earnings per Share (EPS), Quality Assurance (QA), Foreign Ownership (FO), age and sector were extracted using the corporate reports of the chosen companies. Table 3 provides a description of our sample and the dependent variable; IR checklist elements. In addition, independent variables definitions and measurements for the study model are provided in Tables 3 and 4.

---

2    The selected companies' annual reports can be obtained via www.ase.com.

**Table 3.** Research sector—IR elements checklist.

| Sector | No. of. Cases | IR Elements Checklist |
|---|---|---|
| Financial | 38 | 1. Organizational overview and External environment. |
| | | 2. Risk. |
| Manufacturing | 22 | 3. Governance. |
| Services | | 4. Performance. |
| | 22 | 5. Outlook. |
| | | 6. Business Model. |
| Total | 82 | 7. Strategy and resource allocation. |

**Table 4.** Independent variables.

| Variable | Description | Operational Definition | Measurement | Type | Reference |
|---|---|---|---|---|---|
| Size | Size | The total market capitalization expressed in Dinar. | This variable is measured using market value of company's outstanding shares. Market capitalization is equal to the share price multiplied by the number of shares outstanding. | Scale | (Cabral and Mata 2003) |
| EPS | Earnings Per Share: is a very good indicator of the profitability of any organization, and it is one of the most widely used measures of profitability. | Earnings per share (EPS), also called net income per share, is a market prospect ratio that measures the amount of net income earned per share of stock outstanding. Serving as an indicator of the company's financial health. | EPS = net income − preferred dividends average outstanding common shares | Scale | Investopedia |
| Age | Age | Number of years since the firm's foundation. | 2019—First year of foundation. | Scale | (Soliman 2013) |
| QA | Quality Assurance | Quality assurance can be defined as "part of quality management focused on providing confidence that quality Requirements will be fulfilled."[3] | Existence of Quality assurance, such as specific certifications and accreditation (ISO), 0 no QA, 1 if QA exists. | Ordinal (0,1) | (Harvey 2006) |
| FO | Foreign Ownership | Foreign ownership defined as the number of shares owned by noncitizen investors. | This variable has been determined by looking at the ownership section of the annual report of the selected companies and notice whether a company has a foreign investment/ is acquired partially or totally by a foreign individual. 0 no foreign ownership, 1 foreign ownership exists. | Ordinal (0,1) | (Khan et al. 2013; Juhmani 2013) |
| Industry type | Sector | Principal economic activities, a group of similar businesses; services, finance and manufacturing. | Manufacturing 1, services 2 or banking 3 | Ordinal (1,2,3) | |

---

[3]  The confidence provided by quality assurance is twofold—internally to management and externally to customers, government agencies, regulators, and certifiers. An alternate definition is "all the planned and systematic activities implemented within the quality system that can be demonstrated to provide confidence that a product or service will fulfill requirements for quality." American Society for Quality (Anon 2019).

As mentioned before, all financial and other stand-alone reports of the research sample will be examined against the corresponding IR elements checklist listed in Table 3. The next subsection shows the procedures to accomplish the content analysis.

For further explanation, Table 5 shows that a developed weight descriptor assigns discrete weights to (41) items linked to their IR elements with respect to the predefined max scores mapped to each of IR element.

**Table 5.** IR elements and weight descriptor.

| IR Elements | Items | Item Weight Descriptor | Item Max Score | Element Max Score |
|---|---|---|---|---|
| Organizational overview and External environment. | Mission, Vision | 0, no data, 1, mentioned, 2 details. | 2 | 14 |
| | Value and culture | 0, no data, 1, mentioned, 2 details. | 2 | |
| | Ownership and operating structure | 0, no data, 1, mentioned. | 1 | |
| | Principle, market, product, service activities | 0, no data, 1, mentioned, 2 details. | 1 | |
| | Reporting boundary | 0, no data, 1, mentioned. | 1 | |
| | Key quantitative information | 0, no data, 1, mentioned, 2 details. | 2 | |
| | Legal, commercial, social, political, environment | 0, no data, 1, mentioned, 2 details, 3, impact. | 3 | |
| | The number of employees | 0, no data, 1, mentioned. | 1 | |
| | Countries in which the organization operate | 0, no data, 1, mentioned. | 1 | |
| Risk | KPIs mix performance measure | 0, no data, 1, mentioned. | 1 | 2 |
| | KPIs risk indicators | 0, no data, 1, mentioned. | 1 | |
| Governance | Leadership structure, diversity and skill set of those charged with governance | 0, no data, 1, mentioned, 2 details. | 2 | 12 |
| | Action taken to monitor strategic direction | 0, no data, 1, mentioned, 2 details. | 2 | |
| | Reflect of culture values in use of and effect on capitals, relationship with stakeholders | 0, no data, 1, mentioned, 2 details. | 2 | |
| | Compensation policies and plans | 0, no data, 1, mentioned, 2 details. | 2 | |
| | Oversight over the IR process | 0, no data, 1, mentioned, 2 details. | 2 | |
| | Role highest governance body in risk management | 0, no data, 1, mentioned. | 1 | |
| | Role of highest governance body in setting purpose, value and strategy | 0, no data, 1, mentioned. | 1 | |

**Table 5.** *Cont.*

| IR Elements | Items | Item Weight Descriptor | Item Max Score | Element Max Score |
|---|---|---|---|---|
| Performance. | KPIs mix performance measure | 0, no data, 1, mentioned. | 1 | 9 |
| | KPIs risk indicators | 0, no data, 1, mentioned. | 1 | |
| | The organizations effect on the capitals | 0, no data, 1, mentioned. | 1 | |
| | State of key stakeholders' relationship | 0, no data, 1, mentioned. | 1 | |
| | Significant external factors | 0, no data, 1, mentioned. | 1 | |
| | Comparison of actual result vs target | 0, no data, 1, mentioned. | 1 | |
| | Comparison against regional industry benchmarks | 0, no data, 1, mentioned, 2 details. | 2 | |
| | The organization effect positive or negative on the capitals | 0, no data, 1, mentioned. | 1 | |
| Outlook. | Management expectations | 0, no data, 1, mentioned. | 1 | 6 |
| | Likely operating context | 0, no data, 1, mentioned. | 1 | |
| | Uncertainties | 0, no data, 1, mentioned. | 1 | |
| | Real risk with extreme consequences | 0, no data, 1, mentioned. | 1 | |
| | Potential implications | 0, no data, 1, mentioned. | 1 | |
| | Key assumptions | 0, no data, 1, mentioned. | 1 | |
| Business Model. | A simple diagram highlighting key elements by a clear explanation of relevance to organization | 0, no data, 1, mentioned, 2 details. | 2 | 6 |
| | The interdependencies and trade-offs between the six capitals | 0, no data, 1, mentioned, 2 details. | 2 | |
| | Connection to information covered by other content elements, such as strategy. | 0, no data, 1, mentioned, 2 details. | 2 | |
| Strategy and resource allocation. | Short, medium, long term objective | 0, no data, 1, mentioned, 2 details. | 2 | 11 |
| | Implementation plans regarding business model | 0, no data, 1, mentioned, 2 details. | 2 | |
| | Influence from, response to operating context | 0, no data, 1, mentioned, 2 details. | 2 | |
| | Effect on key capitals, risk management arrangement | 0, no data, 1, mentioned, 2 details. | 2 | |
| | Stakeholders consultation in deciding strategies | 0, no data, 1, mentioned, 2 details. | 2 | |
| | An understanding of the organizations ability to adopt to change to achieve goals | 0, no data, 1, mentioned. | 1 | |

*4.2. Descriptive Analysis*

According to the weight descriptor assigned to the IR elements and its evaluation with respect to annual reports, we can find the sum of a certain IR element for each company and then find the actual integrated reporting disclosure score (IRS) using the following equation:

$$IRS = \frac{\sum_i^n RIi}{t} \tag{1}$$

where $i = 1$, $n = 7$ and represent the number of IR elements. $RIi$ = the sum of weighted items. $t$ = max score assigned to an IR element.

*4.3. Impact Analysis*

In an effort to answer the research question, test the proposed hypotheses, and measure the impact extent of corporate characteristics; (size, EPS, QA, FO, age, and sector), on Current Issued Reports (CIRs) conformability, we propose to conduct several statistical tests including correlation, t-test, One-Way-ANOVA and multiple regression.

Therefore, our approach proposes the following null hypothesis that supports answering the second main question and constructing the hypothesized impact model:

**H2.** *There is no significant statistical impact for the corporate characteristics (size, EPS, QA, FO, age, and sector) on the conformability level of CIRs to include content elements of the IR framework.*

Therefore, based on the above hypothesis, the following sub hypotheses are proposed:

**H$_{21}$.** *The size of the company has no significant impact on CIRs.*

**H$_{22}$.** *The age of the company has no significant impact on CIRs.*

**H$_{23}$.** *The EPS of the company has no significant impact on CIRs.*

**H$_{24}$.** *The existence of QA has no significant impact on CIRs.*

**H$_{25}$.** *FO has no significant impact on CIRs.*

**H$_{26}$.** *The sector of the company has no significant impact on CIRs*

Accordingly, the proposed impact can be modeled as follows:

$$CIR = \beta_0 + \beta_1 size + \beta_2 Eps + \beta_3 QA + \beta_4 FO + \beta_5 age + \beta_6 sector + \varepsilon.$$

where $\beta_0$ is a constant, ($\beta_1$ to $\beta_6$) are the regression coefficients (slope) and $\varepsilon$ is the error estimation.

## 5. Results and Discussion

This section presents the statistical results for the proposed model. Statistics describe the conformability extent to which Current Issued Reports (CIRs) conform to the IR elements framework. In addition, it presents the impact of firm's characteristic variables on the extent to which the investigated sectors listed on the ASE comply with the IR framework. This is followed by a comprehensive discussion of the results; conclusion and finally recommendations for future research are provided.

*5.1. Statistical Differences*

As stated earlier, we propose the following first null hypothesis: "There are no significant statistical differences between the conformability level of CIRs to include content elements of the IR framework related to the corporate characteristics (QA, FO, and sector)". Therefore, Pearson correlation, *t*-test, One-Way-ANOVA, and the results are shown in Tables 6 and 7.

**Table 6.** Pearson correlation.

| | Variable | CIR |
|---|---|---|
| QA | Pearson Correlation | 0.543 * |
| | Sig. (2-tailed) | 0.000 |
| FO | Pearson Correlation | 0.115 |
| | Sig. (2-tailed) | 0.304 |
| Sector | Pearson Correlation | −0.163 |
| | Sig. (2-tailed) | 0.143 |

Notes: * $p < 0.01$. b. N = 82.

As shown in Table 6 there is a positive and significant association between CIRs with QA ($\alpha < 0.01$). On the other hand, CIRs have an insignificant association with FO and sector. However, despite correlation results, we still need to check if there is a significant difference between CIRs related to the listed variables. Table 7 shows the result of t-test for both QA and FO; while Table 8 shows the result of One-Way-ANOVA statistical analysis.

**Table 7.** T-test for quality assurance (QA) and foreign ownership (FO).

| | | F | T | Df | Sig. (2-Tailed) | Mean Difference | Std. Error Difference | 95% Confidence Interval of the Difference | |
|---|---|---|---|---|---|---|---|---|---|
| | | | | | | | | Lower | Upper |
| CIR QA | Equal variances assumed | 0.021 | 5.788 | 80 | 0.000 | 0.13820 | 0.02388 | 0.09068 | 0.18572 |
| | Equal variances not assumed | | 5.790 | 75.443 | 0.000 | 0.13820 | 0.02387 | 0.09066 | 0.18575 |
| CIR FO | Equal variance assumed | 0.002 | 1.035 | 80 | 0.304 | 0.02902 | 0.02804 | −0.02679 | 0.08482 |
| | Equal variances not assumed | | 1.035 | 79.990 | 0.304 | 0.02902 | 0.02804 | −0.02679 | 0.08482 |

Based on the results in Table 7, we can simply realize that having QA, or in other words, the existence of a quality control reference would necessarily result in a positive difference (t = 5.79, $\alpha \le 0.05$) in the degree to which CIRs conform to the counterpart's elements of international IR framework. In contrast, FO cannot cause any significant difference (t = −1.035, $\alpha > 0.05$) when companies issue their CIRs.

**Table 8.** One-Way-ANOVA test for Sector.

| CIR | Sum of Squares | Df | Mean Square | F | Sig. |
|---|---|---|---|---|---|
| Between Groups | 0.045 | 2 | 0.023 | 1.415 | 0.249 |
| Within Groups | 1.262 | 79 | 0.016 | | |
| Total | 1.307 | 81 | | | |

Table 8 shows insignificant differences (f = 1.41, $\alpha > 0.05$) in the degree to which CIRs conform to the counterpart's elements of international IR framework related to the sectors (Financial, manufacturing or services).

Based on the results in the Tables 7 and 8, the hypothesis "There are no significant statistical differences between the conformability level of CIR to include content elements of the IR framework related to the corporate characteristics (QA, FO, and sector)" is partially rejected regarding QA and accepted in respect of FO and sector as follows:

- There is a statistically significant difference between the conformability level of CIRs to include content elements of the IR framework related QA.
- There are no statistically significant differences between the conformability level of CIRs to include content elements of the IR framework related to FO and sector.

*5.2. Conformability Results*

The first research question sought to find an answer of "to what extent do Currently Issued Reports (CIRs) of AES listed companies conform to elements of the Integrated Reporting Framework (IRF)?" In different words, we try to find the extent of conformability of CIRs to IR framework.

The descriptive analysis addresses the average score for each individual sector, per subelement of an IR element, compared to the maximum score that can be optimally achieved. Results are shown in Table 9.

**Table 9.** Conformability Results, N = 41 IR elements.

| IR | Sector | Mean | St.d | Max Score | Conformability % | Rank |
|---|---|---|---|---|---|---|
| Organizational overview and External environment. | Financial | 10.42 | 1.98 | 14 | 74% | 2 |
| | Manufacturing | 10.91 | 1.94 | | 78% | 1 |
| | Services | 10.14 | 2.37 | | 72% | 3 |
| Risk | Financial | 1.42 | 0.55 | 2 | 71% | 3 |
| | Manufacturing | 1.59 | 0.50 | | 80% | 1 |
| | Services | 1.50 | 0.50 | | 75% | 2 |
| Governance | Financial | 8.16 | 0.68 | 12 | 82% | 1 |
| | Manufacturing | 7.05 | 1.81 | | 70% | 3 |
| | Services | 7.09 | 1.51 | | 71% | 2 |
| Performance | Financial | 6.87 | 1.32 | 9 | 57% | 1 |
| | Manufacturing | 6.36 | 1.29 | | 53% | 2 |
| | Services | 6.41 | 0.91 | | 53% | 3 |
| Outlook | Financial | 3.16 | 1.52 | 6 | 53% | 2 |
| | Manufacturing | 3.41 | 1.68 | | 57% | 1 |
| | Services | 2.55 | 1.09 | | 42% | 3 |
| Business | Financial | 2.63 | 1.07 | 6 | 53% | 1 |
| | Manufacturing | 2.50 | 1.14 | | 50% | 2 |
| | Services | 2.27 | 1.08 | | 45% | 3 |
| Strategy | Financial | 5.05 | 1.89 | 11 | 56% | 1 |
| | Manufacturing | 4.59 | 1.40 | | 51% | 2 |
| | Services | 4.45 | 1.26 | | 49% | 3 |

Referring to Table 9 the conformability result shows a noticeable variance among reported IR elements ranging between (0.42) and (0.82). Although none of our sample has yet published an integrated report, there were many companies with high reporting scores whose reports included approximately most of the content elements required for an integrated reporting framework. On the other hand, some companies obtained relatively low scores, particularly in the services sector.

As presented in the Table, the "Governance" element of the financial reports ranked one with (82%) compared to the manufacturing and services sectors. while the "outlook" element of services sector indicates the lowest conformability level with (42%) among other sectors and among IR elements. Likewise, Table 4 shows the remaining conformability of the same sectors and the rank

for each associated IR element. Initially, we need to note that the service sector mostly shows a low level of conformability compared to the financial and manufacturing sectors. Figure 2 summarizes conformability level by sector.

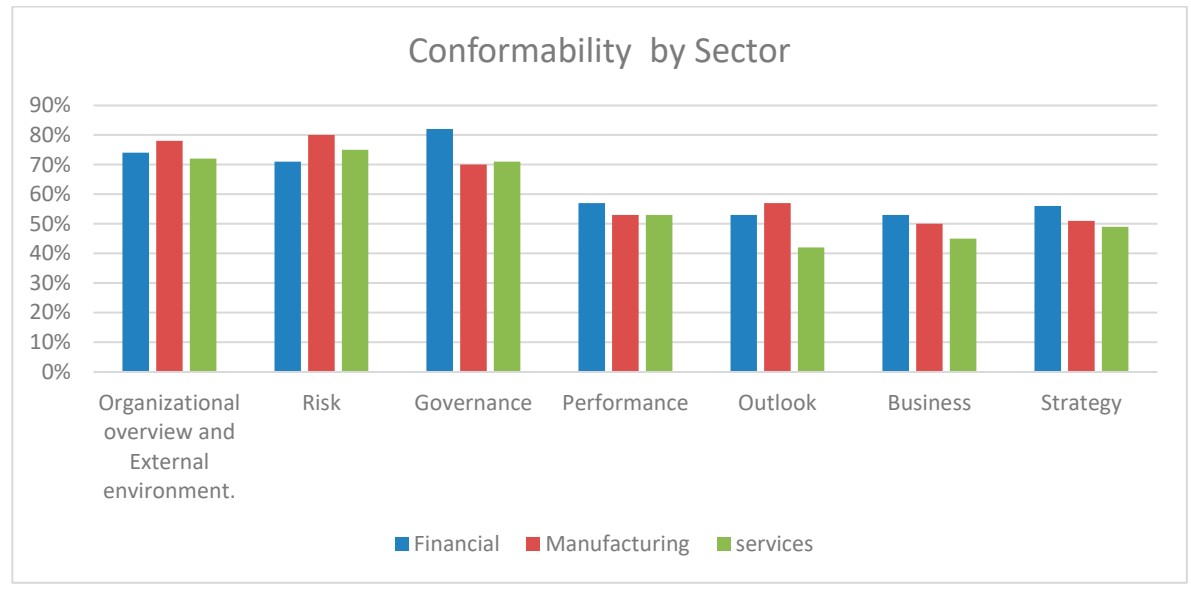

**Figure 2.** Conformability by sector.

Regarding the organizational overview and external environment elements, the results show that the financial, manufacturing and services sectors maintain (74%), (78%) and (72%) of CIRs conformability level, respectively. CIRs related to the risk element achieve (71%), (80%) and (75%) of CIR conformability in the financial, manufacturing and services sectors respectively. For governance elements, statistics show that the financial sector is the most conformable sector with 82% of the IR elements framework, while the manufacturing sector has 70% of the IR elements framework. The performance element included in CIRs shows an approximate level of conformability among different sectors to reach 57% in favor of the financial sector. Similarly, the reported outlook element reaches 57%, but in favor of the manufacturing sector. The sixth IR element, namely business, best conforms at 53% of the financial sector. Finally, the strategy element conforms to 56% of the financial sector. Based on the aforementioned results, the answer of the first research question is: CIRs of all sectors conform to the IR content elements at 62%.

Again, Table 10 answers the first research question and reveals that the service sector shows the lowest conformability level compared to other sectors, whereas the financial firms conform at 65%, which is still unsatisfactory in terms of the IR elements framework.

**Table 10.** Current Report Performance.

|  | Sector | Mean | Std. Deviation | Conformability % |
|---|---|---|---|---|
|  | Financial | 37.71 | 7.16 | 65.0% |
| CIR | Manufacturing | 36.40 | 7.88 | 62.8% |
|  | Services | 34.40 | 7.03 | 59% |

This result, which is still insufficient but better than the services sector, might be because sectors that mainly depend on scarce natural resources are motivated to report further issues of social and environment, to represent a better understanding of its social responsibility, and the connection between financial and nonfinancial performance (Eccles and Armbrester 2011; Ernst and Young 2014). Also, to appear capable of creating value in the future, to attain legitimacy, and to be recognized as different within the sector in a way that society perceives them to be more favorable (Suchman 1995).

### 5.3. Impact Analysis

Central to the research hypothesis, which states, "There is no significant statistical impact for the corporate characteristics (size, EPS, QA, FO, age, and sector) on the conformability level of CIRs to include content elements of the IR framework," the multivariate statistical analysis using multiple regression was performed. Table 11 shows the results of the multiple regression analysis.

**Table 11.** Multiple Regression Analysis.

| Model | | R | R$^2$ | F | Sig | Unstandardized Coefficients | | Standardized Coefficients | t | Sig. |
|---|---|---|---|---|---|---|---|---|---|---|
| | | | | | | B | Std. Error | Beta | | |
| | (Constant) | | | | | 0.483 | 0.042 | | 11.479 | 0.000 |
| | Sector | | | | | 0.007 | 0.013 | 0.044 | 0.498 | 0.620 |
| | Size | | | | | $5.270 \times 10^{-10}$ | 0.000 | 0.392 | 4.365 | 0.000 |
| 1 | EPS | 0.74 | 0.55 | 15.06 | 0.000 | $5.753 \times 10^{-9}$ | 0.000 | 0.001 | 0.008 | 0.994 |
| | Age | | | | | 0.002 | 0.001 | 0.257 | 2.799 | 0.007 |
| | QA | | | | | 0.093 | 0.021 | 0.366 | 4.347 | 0.000 |
| | FO | | | | | −0.002 | 0.021 | −0.007 | −0.081 | 0.935 |

The proposed hypothetical models were investigated with multi-regression analysis. Table 11 shows the impact of the sector, size, EPS, Age, QA and FO on the CIR. Model 1 shows that size, EPS, QA, FO, age and sector explain 55% of the conformability variance represented by the value of R2. Thus, the F value proves to be a significant impact for these variables on CIRs (f = 15.06, α ≤ 0.05). Therefore, we reject the null hypothesis and accept the alternative, which states that "There is a significant statistical impact for the corporate characteristics (size, EPS, QA, FO, age and sector) on the conformability level of CIR to include content elements of the IR framework". However, model 1 shows that sector, EPS and FO individually have no impact on CIR. Accordingly, a linear regression was applied to precisely determine the impact of each variable on CIRs. Table 12 shows the results.

**Table 12.** Simple Linear Regression.

| Model | R | R$^2$ | F | Unstandardized Coefficients | | Standardized Coefficients | T | Sig. |
|---|---|---|---|---|---|---|---|---|
| | | | | B | Std. Error | Beta | | |
| (Constant) | 0.19 | 0.03 | 2.88 | 0.679 | 0.033 | | 20.522 | 0.000 |
| Sector | | | | −0.028 | 0.017 | −0.185 | −1.681 | 0.097 |
| (Constant) | 0.6 | 0.36 | 0.45 | 0.581 | 0.013 | | 43.621 | 0.000 |
| Size | | | | $8.078 \times 10^{-10}$ | 0.000 | 0.601 | 6.734 | 0.000 |
| (Constant) | 0.5 | 0.03 | 0.23 | 0.628 | 0.014 | | 44.294 | 0.000 |
| EPS | | | | $-5.100 \times 10^{-7}$ | 0.000 | −0.053 | −0.479 | 0.633 |
| (Constant) | 0.48 | 0.22 | 22.7 | 0.515 | 0.027 | | 19.060 | 0.000 |
| Age | | | | 0.003 | 0.001 | 0.470 | 4.765 | 0.000 |
| (Constant) | 0.54 | 0.3 | 33.5 | 0.568 | 0.016 | | 35.914 | 0.000 |
| QA | | | | 0.138 | 0.024 | 0.543 | 5.788 | 0.000 |
| (Constant) | 0.16 | 0.01 | 1.07 | 0.614 | 0.020 | | 30.984 | 0.000 |
| FO | | | | 0.029 | 0.028 | 0.115 | 1.035 | 0.304 |

Results in Table 12 disclose that the sector variable can explain only 3% of the variance of CIR including IR elements (f = 2.88, α > 0.05). Company size has a significant impact on CIR and explains

36% of the variance of CIR including IR elements (f = 0.45, $\alpha \leq 0.05$). EPS has no significant impact and can only explain 3% of the variance of CIR including IR elements (f = 023, $\alpha > 0.05$). As for age, it has a significant impact on CIR and explains 22% of CIR including IR elements (f = 22.7, $\alpha \leq 0.05$). QA also maintains a significant impact on CIR and explains 30% of variance of CIR including IR elements (f = 33.5, $\alpha \leq 0.05$). Finally, FO has no significant impact to explain an influential portion of variance of CIR including IR elements (f = 1.07, $\alpha > 0.05$). Therefore, the decision for subhypotheses related to H2 can be finalized as shown in Table 13.

**Table 13.** Subhypotheses final decision.

| Subhypotheses | Decision | Argument | Alternative |
|---|---|---|---|
| $H_{21}$: The size of the company has no significant impact on CIRs | Rejected | Significant at ($\alpha \leq 0.05$) | H21: The size of the company has a positive impact on CIRs. |
| $H_{22}$: The age of the company has no significant impact on CIRs. | Rejected | Significant at ($\alpha \leq 0.05$) | H22: The age of the company has a positive impact on CIRs. |
| $H_{23}$: The EPS of the company has no significant impact on CIRs | Accepted | Insignificant at ($\alpha \leq 0.05$) | |
| $H_{24}$: The existence of QA has no significant impact on CIR. | Rejected | Significant at ($\alpha \leq 0.05$) | H24: The existence of QA has a positive impact on CIRs. |
| $H_{25}$: FO has no significant impact on CIRs. | Accepted | Insignificant at ($\alpha \leq 0.05$) | |
| $H_{26}$: The sector of the company has no significant impact on CIRs. | Accepted | Insignificant at ($\alpha \leq 0.05$) | |

Table 13 shows the individual impact for each independent variable and its explanatory capacity of CIR through the simple linear regression model. Yet, in comparison with the model produced in Table 12, it shows the possibility of eliminating noninfluential variables (EPS, FO, and sector) to construct a new model to reduce the number of independent variables in order to simplify the regression model. Table 14 shows the impact size of influential variables.

**Table 14.** Influential Multiple Regression Analysis.

| | Model | R | $R^2$ | F | Sig | Unstandardized Coefficients | | Standardized Coefficients | T | Sig. |
|---|---|---|---|---|---|---|---|---|---|---|
| | | | | | | B | Std. Error | Beta | | |
| | (Constant) | | | | | 0.499 | 0.022 | | 23.096 | 0.000 |
| 2 | Size | 0.74 | 0.55 | 31.11 | 0.000 | $5.203 \times 10^{-10}$ | 0.000 | 0.387 | 4.493 | 0.000 |
| | Age | | | | | 0.002 | 0.001 | 0.242 | 2.906 | 0.005 |
| | QA | | | | | 0.093 | 0.021 | 0.364 | 4.482 | 0.000 |

The multiple regression analysis in Table 14 shows that model 2 can interpret 55% of the variance of CIR including IR elements (f = 31.11, $\alpha \leq 0.05$). This is consistent with that which is stated in Table 12 and confirms the inability of the company sector, EPS and FO to improve the explanatory level. Thus, this research adopts the following simplified model to explain 55% of variance of CIR including IR elements as follows:

$$CIR = \beta_0 + \beta_1 size + \beta_3 QA + \beta_5 age + \varepsilon.$$

The results indicate that there are significant statistical differences related to the size, age, and QA of firms, while there are no significant statistical differences related to the EPS, sector, and foreign-ownership. This complements the findings of Haniffa and Cooke (2002), and Eng and Mak (2003), who found no significant association between firm ownership and voluntary disclosure level.

In addition, the results of the multiple regression analyses indicate the significance of these differences and their ability to explain (55%) of the variance in the conformability level.

Many previous studies are concerned with corporate traditional and social responsibility reports that have supported the results of our study. Therefore, the findings of these previous studies can be useful for explaining our results, as IR is a type of advanced corporate reporting that includes both financial and nonfinancial information, but also has the philosophy of integrated thinking and capitals dependency. Nevertheless, the literature has shown that there is an argument about the purposes of firms providing accountability and voluntary disclosures, some assigning it to the concept of firms acting in the stakeholders' best interest as illustrated by the stakeholder theory. Other studies assigned it to an effort to gain legitimacy for the firm as described by legitimacy theory (Campbell 2007; Rowbottom and Locke 2014; Beck et al. 2017).

Referring to our result of the present study, particularly the variable of size, studies such as (Hartikayanti et al. 2016; Andrikopoulos and Kriklani 2013; Barako et al. 2006; Milanés-Montero and Pérez-Calderón 2011; Wallace et al. 1994; Eng and Mak 2003; Alsaeed 2006; Chau and Gray 2010), have found a positive association between firm size and voluntary disclosure level and corporate reporting.

The study of Singhvi and Desai (1971) argues that larger firms tend to disclose more information, as the accumulation, processing and disclosure cost of information is not high compared to smaller companies. In addition, they found that management of larger firms consider the likely benefits of disclosing more information, such as greater reputation, marketability and better ease of financing.

The possible other reason behind our result is because large firms are more exposed to the public than small firms, are more complex and affect a wide range of society, thus, they are persuaded to disclose more information (Patten 2002; Cormier and Gordon 2001; Alsaeed 2006; Cooke 1989).

Another likely reason stated by (Galani et al. 2011) is that larger firms care about their reputation. Thus, they try to increase their transparency to gain public trust, and preserve their position in society, through enhancing the quality of their reporting, because they believe the better their reports are, the better society and stakeholders will recognize them, as explained by legitimacy and stakeholder theories (Ching and Gerab 2017).

Legitimacy and stakeholder theories can add to our discussion, that larger companies have good-quality disclosure and more information compared to smaller ones, that "this should enhance (an organization's) legitimacy with groups of stakeholders (social, economic and environmental audiences) in meeting their specific needs and regulatory expectations" (Ching and Gerab 2017, p. 100) and that this would strengthen stakeholders' trust (Ching and Gerab 2017).

Similarly, the result of the study regarding the impact of the firm's age, states that the age of a company has a positive significant impact on the conformability level of CIRs to include content elements of the IR framework. This is supported by previous studies (Hossain and Hammami 2009; Owusu-Ansah 1998). Although those studies did not investigated IR, they investigated the association between firm age and disclosure level (traditional reporting and sustainability and social reports), but their results can be used to explain our phenomenon, as IR is a type of corporate reporting as mentioned before. The study of Owusu-Ansah (1998) argues that younger companies may encounter problems of collecting, processing, and disclosing information, which may make it a more costly and difficult process. In the same way, Liu and Anbumozhi (2009) clarify that if a firm is established a long time ago, it would be a mature experienced business that has steady customers and more saturation, this requires that a company should provide more financial and nonfinancial information to satisfy its stakeholders and meet its financial, social, and environmental obligations.

In view of these previous studies, we suggest that older companies better understand what information should be disclosed in the reports, and this will enable them to report information of high quality and hence will increase the adherence level of the company's disclosures to the IR framework.

Finally, the result of the impact of quality assurance (and there is a clear lack of studies when deciding its impact on the IR arena particularly, and on corporate disclosure in general) indicates that QA has a significant positive impact on the conformability level of CIRs. It enhances the extent

and quality of voluntary environmental disclosures, and hence will increase the adherence level of company disclosures to the IR framework. In other words, the study finds that companies that obey specific external assurance and professional examination, such as the International Organization for Standardization (ISO), have more of IR elements, and disclose more nonfinancial information compared to nonassured companies. This result is compatible with a study of (Perego and Kolk 2012) which stated that assurance has a significant role in improving the quality of CSR reports.

To clarify this more, the QA strategy focuses on minimizing the negative environmental impact of a firms' products throughout their development, and improving corporate environmental performance, such as energy consumption and air pollution (Crane et al. 2016). This resulted in benefits for the stakeholders involved and for the environment by different means, such as publishing more financial and nonfinancial disclosure[4]. In arriving at this conclusion, the study found that conformability level of CIRs to include content elements of IR is impacted positively by QA.

In summary, quality assurance focuses on sustainability and social–environmental issues. Therefore, companies that adopt ISO and other quality management certifications will give a lion-share for the sustainability aspect and this will cover many issues that IR is concerned with, and this should increase the conformability of companies that report to the IR framework. Figure 3 shows the regression plot that justifies the impact of size, age, and QA.

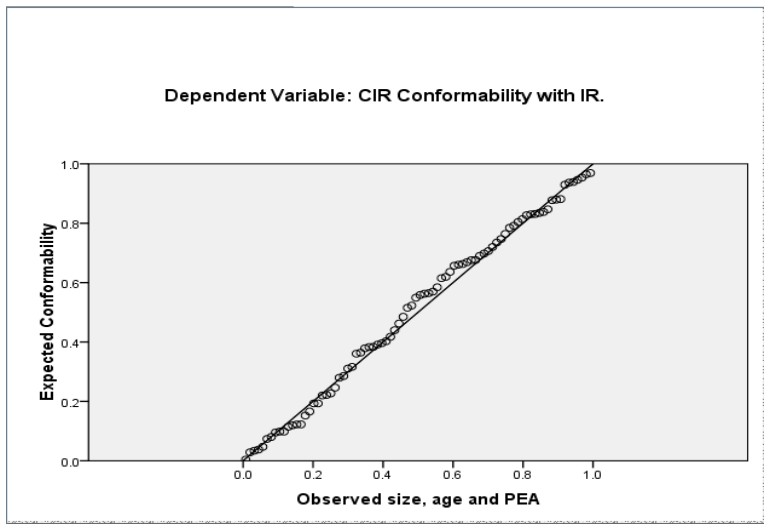

**Figure 3.** Current Issued Reports (CIR) conformity with IR.

To conclude, this study revealed the existence of differences among our sectors to report IR elements within its reports. Results indicate that if a company is older, larger and has a quality assurance the likelihood of disclosing more elements of IR increases.

The firm's EPS, foreign ownership, and type of industry have no significant impact on corporate conformability disclosure with IR framework.

## 6. Conclusions

This study has contributed to IR research, which, as a field, has previously received very little recognition among scholars in Jordan. Moreover, IR still does not exist in Jordan's business practices.

---

4    "A number of alternative frameworks for sustainability and environmental reporting have appeared in recent years, typically with a stronger focus on reporting primarily for investors. The most notable ones are "integrated reporting" which focus on a framework for reporting sustainability and the sustainability accounting standards developed by the Sustainability Accounting Standards Board (SASB), which aim to help public corporations disclose material, decision useful information to investors" (Crane et al. 2016).

Principally, the objectives of this study were to determine the level of conformity between CIRs and IR elements of the ASE listed companies, as well as to determine whether the investigated corporate characteristics (size, age, QA, EPS, industry type, FO) of these companies had any impact on the conformability of Current Issued Reports (CIRs).

Based on adoption of the IR framework and using the method of content analysis, corporate annual reports and other stand-alone reports of 82 companies in 2017 and 2018 within the financial, industrial, and services sectors, were chosen for this study.

The findings of the study provide an answer to the research question and show that sectors vary in their levels of conformability. It reveals that the service sector shows the lowest conformability compared to other sectors, whereas the financial firms conform 65% followed by the industrial sector. It also finds a positive association between CIR conformability and variables of size, age of company and quality assurance. However, EPS, FO and type of industry were found to have no impact on the conformability of CIRs to the IR framework.

## 7. Limitations

Although a reasonable sample size is considered in this study, there is a need in future studies to enlarge sample sizes to include all listed companies in the ASE. The study has investigated specific variables, while others have not been covered. Thus, future research should take those variable corporate characteristics into consideration in order to provide a comprehensive answer to the research questions.

## 8. Recommendations and Future Work

As the body of literature on integrated reporting continues to grow, further research is necessary to extend the existing knowledge of IR.

1. Future research may seek to capture the perspective of decision-makers in Jordan towards the adoption of IR.
2. Future studies may address the link between IR and other variable corporate characteristics that are not captured in this study and might affect the adoption of IR in the future.
3. Future research may use different research methods and approaches to investigate variations in levels of conformability.
4. This study recommends future work to investigate the need to modify accounting curricula in Jordan in order to provide students with up-to-date knowledge and information on various approaches to corporate communication and reporting.
5. Companies are recommended to pay more attention to conforming their CIR to the IR framework.

**Supplementary Materials:** All empirical research data were obtained from Amman Stock Exchange, and the data are available online at https://www.ase.com.jo/en.

**Author Contributions:** Conceptualization, G.A.A.; methodology, G.A.A. and A.O.A.-H.; software, G.A.A. and A.O.A.-H.; validation, G.A.A. and A.O.A.-H.; formal analysis, G.A.A. and A.O.A.-H.; investigation, A.O.A.-H.; resources, A.O.A.-H.; data curation, A.O.A.-H.; writing—original draft preparation, G.A.A. and A.O.A.-H.; writing—review and editing, G.A.A.; visualization, A.O.A.-H.; supervision, G.A.A.; project administration, G.A.A. All authors have read and agreed to the published version of the manuscript.

**Funding:** This research received no external funding.

**Conflicts of Interest:** The authors declare no conflict of interest.

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
