# Peer review of "Conformity of Annual Reports to an Integrated Reporting Framework: ASE Listed Companies"

_ijfs, doi:10.3390/ijfs8030050_

Round 1

Reviewer 1 Report

IJFS-857955

International Journal of Financial Studies

Conformity of annual reports to an integrated reporting framework: ASE listed companies

Referee Report

The authors examine the level of conformity between issued financial reports of firms listed on the ASE and integrated reporting (IR) elements proposed by the IIRC. They also study whether corporate characteristics such as firm size, age, profitability, and industry have an impacting effect on such conformity. They find a variation in the levels of conformity across different industry sectors, with the financial sector exhibiting the highest level and the service sector showing the lowest, with the industrial sector in between. The size, age, and quality assurance of a firm are also positively associated with conformity.

I commend the authors on taking a first step into a topic that is novel to not only Jordanian markets, but also many other places in the world. The paper is very well-written. I see their idea as an interesting starting point. I do, however, have some suggestions that I would like to share with them.

The motivation behind the paper, as the authors state, is to stress the importance of adopting IR and understanding where Jordanian firms stand. Under this principle, I do not feel that the authors are delivering the first and probably more important contribution. For example, all else equal, are firms with higher IR-conformity priced more accurately? Do they enjoy lower industry-adjusted costs of capital? One way or the other, I hope to see that IR adoption leads to results that can somehow be materialized.

Further, I feel that there are parts of the paper left unexplained. Viewing the results, are financial firms more heavily regulated in Jordan and therefore being self-pushed to conform more to IR? Does quality assurance a driving force behind adopting IR, or is it really the reverse? The authors include some variables that had not been examined in prior literature. What are the economic reasons behind each of these variables? As a reader, I was curious about all this, but I cannot find the answers in the paper.

Analytically, I also think that more can be done to ensure the validity of the authors’ arguments, so that their idea may come across more convincing to readers and, perhaps more importantly, policymakers. For instance, why is the sample period limited to only 2017 and 2018? Why is the study limited to only 82 firms in three industries? Of course, expanding the sample depends largely on whether the content analysis was conducted through hand collection or some other machine-readable programming.

Presentation-wise, some of the variables that the authors claim as a novelty for the literature produce non-results. These can simply be placed in a short discussion section in the paper; I really do not see the point of highlighting them as contributions. In fact, they can potentially distract a reader’s attention on other more interesting results presented by the authors. There are also multiple instances in the paper where words are misspelled and where the writing is missing punctuations or grammatically incorrect. Sometimes there seem to be digits coming out of nowhere. The authors should also make sure that their citations are consistent throughout the paper. Some of the variables, such as IR/QA are not made clear of their definition at first appearance; these can be easily fixed for reader-friendliness.

Good luck!

Author Response

Could you please see the attached document?

Reviewer 2 Report

The authors of the article consider the urgent problem of integrated enterprise reporting. Methodological aspects, techniques and reporting indicators of financial and non-financial reporting are indeed a serious practical problem, especially when benchmarking is necessary.
As for the research presented in the article, it would be helpful if the authors could clarify the following:

1) why the sample size is different for the financial sector and how the sample size can affect the results (table 3);

2) the difference between the elements and the categories of the Integrated reporting (41 elements are suggested by the text, and 7 by the formula (1));

3) how were the results of table 5 obtained and was formula (1) used for this? How was the input element t=max score obtained?

4) why was only the second hypothesis considered?

5) how can the results help improve integrated reporting practices (particularly in Jordan)?

Author Response

(The authors gave the same response as above.)

Reviewer 3 Report

General Comment:

This is an interesting article, that is contributing to the very large discussions on sustainable finance and integrated reporting, something that has gained of influence in the recent years. This contribution is focusing on Jordan companies, that, as the authors explain, are at an early stage of maturity in this area. Therefore, this is an interesting contribution.

The Abstract is OK but you use acronyms that are not defined; therefore you make the readability markedly less friendly. As a  general comment: when you introduce acronyms, you should be careful at explaining them at the very beginning. Acronyms such as ASE, FO, etc…. are used at the beginning but are only explained later in the text.

In terms of language, the sections 1 to 3 are OK, but the 2nd part would deserve a polishing, for clarity and style.

More specifically, below are some comments related directly to your text, that might help you to increase the impact of your article.

To 1: Introduction: good structure, with comprehensive coverage. Some points that might increase impact:

The issues related to non-financial reporting are pretty well covered. It could have more impact if it would be related to the on-going discussions on sustainable finance. To cut it short, non-financial reporting does matter because:

  • The marktet value of companies tend to embed more and more non-tangible assets, such as image, IP, relationship, etc…, all things that may be captured in a non-financial reporting, but certainly not in a financial report. The value chain model of the IIRC can support the approach.
  • Non-financial reporting is facing issues of non-reproducibility (i.e. various rating agencies can have very different ratings), and this is impacting the credibility of the whole idea. Reasons for this are often linked to:
    • A lack of standardized methodologies
    • Issues with the databases related to ESG impact rating
    • And also an issue with the raters of the rating agencies, as this is often done in a rather subjective way.
  • The introduction of the article points as well to the shortcomings of CIF reports, but it would be worth expanding on this, how such reportings come short to societal expectations, and how IR can bridge this gap.

The authors relate quite well the methodologies to the IIRC

To 2: what is IR

When you elaborate on IR, in relation to the framework of IIRC, any reason you are not linking this to the value creation model of the IIRC, as, after all, this is at the core of the purpose of IR, to measure the whole value creation process.

By the way, IIRC has just issued a revision of its framework, and this should be reflected in this section. Your text refers to a framework and guiding principles of IIRC that are now obsolete.

Also: are you sure you want to have the IR content listing in the main text?

To 3: Literature Review

Good coverage of IR, with issues of predictability of performance, of sustainability/environmental impact being typically the bulk.

L-196: what do you understand with user's sophistication?

L-229+ff: this seems to be rather related to the research methodology??

L-239-251: could you try to polish the text? I think this would gain on readability.

To 4: Research Methodology

As content analysis is a very large topic, having even roots to the mathematical coding theory, with software commonly used for text analysis, your text would gain of impact by elaborating on the tools and methodology used. Your text tends to replicate statements at the same level without adding clarifications. E.g.: L-273, with 41 elements, and L-297.

The para 4.1. should be revisited, for the phrasing and the content.

You could elaborate on the reasons you have selected the independent variables. E.g.: QA may have different levels, from ISO-9k certification to full-fledge Management systems in place, which might be also relevant for an IR reporting. F.O: could indicate a level of foreign dependency. Is there a connection with the IIRC framework? If yes, you may wish to elaborate.

L-323: explain acronym PEA

You refer to testing H02, not dealing with H01. No problem, but then, you may wish to simplify lines L-247 to L-251.

L-333: this equation brings clarity to your approach.

To 5: Results and Discussion

I personally have difficulties to trace the process that drives to Table 5: how do you get to these figures, especially conformability. May be your text could gain on impact by elaborating on the pathway. This could add credibility to your statements L-355+ff. Especially, how you quantify Organizational…., Risk, Governance, etc….

L-377-379: elaborate and clarify.

L-418: you mean table 9, or 4.9?

P.16: Interesting discussion on the rationality of size vs. disclosing level, well-anchored into conclusions from other publications.

P.17, the discussion on QA could be extended. Firms that have an ISO-9k, 14k, and 18k have tools in place to measure their impacts, and therefore to improve them. Once here, they may want to communicate about it. But firms that do not have this may not be able to measure their impact, and therefore will not be able to communicate beyond greenwashing. Even though the discussion is more complex….

Also: specify what you mean with QA: along with ISO 9k, this is centered on consumer satisfaction, while ISO-14k relates to the environment. And the social impact goes beyond.

L-501: provision of external assurance: please clarify what you mean. Is this about an external certification?

L-510: see also above. Quality assurance is part of an integrated management system that covers environment and social impact.

To 6: Conclusions

L-532: for better readability, you may want to state again the “research question”.

To 7: Limitations:  no comment.

To 8: Recommendations:

L-549: 2.?????

You may wish to beef-up paras 6 to 8.

Author Response

(The authors gave the same response as above.)

Round 2

Reviewer 1 Report

IJFS-857955.R1

International Journal of Financial Studies

Conformity of annual reports to an integrated reporting framework: ASE listed companies

Referee Report

The authors examine the level of conformity between issued financial reports of firms listed on the ASE and integrated reporting (IR) elements proposed by the IIRC. They also study whether corporate characteristics such as firm size, age, profitability, and industry have an impacting effect on such conformity. They find a variation in the levels of conformity across different industry sectors, with the financial sector exhibiting the highest level and the service sector showing the lowest, with the industrial sector in between. The size, age, and quality assurance of a firm are also positively associated with conformity.

In the previous round, I brought up that the motivation behind the paper, as the authors state, is to stress the importance of adopting IR and understanding where Jordanian firms stand. Under that principle, I did not feel that the authors are delivering the first and probably more important contribution. I hoped to see that IR adoption leads to results that can somehow be materialized.

I read the authors’ response. While I understand that page limitation is a concern and am perfectly fine with the authors not conducting additional analyses to address the concern, I suggest that the authors (i) cite alternative sources, such as news articles and/or other marginally-related research, to at least qualitatively discuss the potential importance of IR and (ii) emphasize the contribution of the paper as providing an understanding of the whereabouts of Jordanian firms. The authors may pose the task of quantitatively materializing the importance of IR as a venue for future research. Doing so explicitly provides inspiration for follow-up work and potentially elevates the impact of IJFS.

Thank you and good luck!

Author Response

Reviewer 1

Reviewer In the previous round, I brought up that the motivation behind the paper, as the authors state, is to stress the importance of adopting IR and understanding where Jordanian firms stand. Under that principle, I did not feel that the authors are delivering the first and probably more important contribution. I hoped to see that IR adoption leads to results that can somehow be materialized.

Response:

Dear reviewer, thank you very much for your valuable point. It should be investigated in the near future as it raise my curiosity to research it.

And I am Sorry, I think there is a misinterpreted regarding this point, thus, I removed it from the paper.  Nevertheless, the main objectives and motivation of this research are to determine the level of conformity between Current Issued Reports (CIRs) and Integrated Report (IR) elements of the Amman Stocks Exchange (ASE) listed companies, as well as to determine whether the investigated corporate characteristics (size, age, quality assurance (QA), earning per share (EPS), industry type, foreign ownership (FO)) of these companies have any impact on the conformability of CIRs. Thus, this paper has two main questions:

  • To what extent do Currently Issued Reports (CIRs) of AES listed companies conform to elements of the Integrated Reporting Framework?
  • What is the impact of corporate characteristics (Size, EPS, quality assurance, foreign ownership, and sector) on the levels of conformity of CIRs to include IR elements in companies’ reports?

And when I stated “to stress the importance of adopting IR” I did not mean to examine what is the benefit or importance of IR if Jordanian companies adopt it, (however, your point still a very important and interesting)

Actually, What I meant here, is to start investigating and researching this important and absent topic within Jordanian context, and doing this would raise the awareness of all stakeholders including capital market, financial statements users, academics and professionals of the need to adopt such approach, and also would encourage other to research this topic more and more  

The study makes a contribution in several ways; it is the first to approach this research within the context of Jordan. It serves as a starting point for future studies, and may contribute to changing the way companies disclose their information. In addition, this study makes a contribution by providing an initial assessment and practical vision on where CIRs of ASE stands with regard to IR requirements. This study contributes to the literature by responding to various recent calls in the area of IR. It also brings insight from developing countries, where little is known about this topic

Thank you again

Reviewer 2 Report

  1. It is recommended that the text of the article more clearly explain formula (1) and the corresponding results presented in table 8 (Conformability Results, N=41 IR elements).

2. Table 3. Research sample and Input - IR elements checklist

Do you prefer to use the definition of "elements"? Or, as you pointed out in the response "The employed disclosure index included 41 elements within
seven categories", could “categories” be used?

Author Response

Reviewer: 2): It is recommended that the text of the article more clearly explain formula (1) and the corresponding results presented in table 8 (Conformability Results, N=41 IR elements).

Response: sorry, I am not sure what do you mean exactly,

However,  I provided an explanation of what I think you consider it an important and should be  added to make the article more explicit    (highlighted L. 313, table to 317)

Reviewer: 2: Table 3. Research sample and Input - IR elements checklist (done)

Thank you
